# pH-Sensitive Nanoparticles for Colonic Delivery Anti-miR-301a in Mouse Models of Inflammatory Bowel Diseases

**DOI:** 10.3390/nano13202797

**Published:** 2023-10-20

**Authors:** Junshan Wang, Min Yao, Jiafeng Zou, Wenxing Ding, Mingyue Sun, Ying Zhuge, Feng Gao

**Affiliations:** 1Department of Gastroenterology, Chongming Branch of Shanghai Tenth People’s Hospital, Tongji University School of Medicine, Shanghai 202157, China; 2Shanghai Frontier Science Research Base of Optogenetic Techniques for Cell Metabolism, School of Pharmacy, East China University of Science and Technology, Shanghai 200237, China; ym1584007995@163.com (M.Y.); zoujiafeng2019@126.com (J.Z.); dwx150120413@gmail.com (W.D.); 17854280576@163.com (M.S.); 3Department of Cardiology, Shanghai General Hospital, Shanghai Jiao Tong University School of Medicine, Shanghai 200025, China; yiliaobangzu@163.com; 4Shanghai Key Laboratory of New Drug Design, School of Pharmacy, East China University of Science and Technology, Shanghai 200237, China; 5Shanghai Key Laboratory of Functional Materials Chemistry, East China University of Science and Technology, Shanghai 200237, China; 6Optogenetics and Synthetic Biology Interdisciplinary Research Center, State Key Laboratory of Bioreactor Engineering, East China University of Science and Technology, Shanghai 200237, China

**Keywords:** anti-miR-301a, oral nanoparticle delivery system, colon targeting, pH-responsive, inflammatory bowel disease

## Abstract

Though the anti-miR-301a (anti-miR) is a promising treatment strategy for inflammatory bowel disease (IBD), the degradability and the poor targeting of the intestine are a familiar issue. This study aimed to develop a multifunctional oral nanoparticle delivery system loaded with anti-miR for improving the targeting ability and the therapeutic efficacy. The HA-CS/ES100/PLGA nanoparticles (HCeP NPs) were prepared using poly (lactic-co-glycolic acid) copolymer (PLGA), enteric material Eudragit^®^S100 (ES100), chitosan (CS), and hyaluronic acid (HA). The toxicity of nanoparticles was investigated via the Cell Counting Kit-8, and the cellular uptake and inflammatory factors of nanoparticles were further studied. Moreover, we documented the colon targeting and pharmacodynamic properties of nanoparticles. The nanoparticles with uniform particle size exhibited pH-sensitive release, favorable gene protection, and storage stability. Cytology experiments showed that anti-miR@HCeP NPs improved the cellular uptake through HA and reduced pro-inflammatory factors. Administering anti-miR@HCeP NPs orally to IBD mice markedly reduced their pro-inflammatory factors levels and disease activity indices. We also confirmed that anti-miR@HCeP NPs mostly accumulated in the colon site, and effectively repaired the intestinal barrier, as well as relieved intestinal inflammation. The above nanoparticle is a candidate of the treatment for IBD due to its anti-inflammatory properties.

## 1. Introduction

Inflammatory bowel diseases (IBDs), carrying a very high risk of colon cancer, are high-morbidity chronic idiopathic disorders occurring in the gastrointestinal tract (GIT). The number of IBD patients has steadily increased, especially in developed countries [1,2]. Recent studies have associated the etiology of IBD with a genetic susceptibility, an inordinate immune response, a damaged mucosal barrier, and an abnormal gut microbiota [3,4]. Pharmacotherapies have been widely applied clinically [3,5,6]. However, these medications have poor targeting abilities and cause severe adverse effects [5,7,8]. Meanwhile, herbal ingredients have been widely used as anti-inflammatory drugs [9,10,11]. In our previous study, we delivered shikonin to the colon for the treatment of IBD, but the mechanism was still vague due to the multi-target properties of herbal ingredients [9,11]. Nowadays, microRNAs (miRNAs) with a precise mechanism have attracted much attention. miRNAs are a class of small, endogenous, non-coding RNA that regulate gene expression and play critical roles in various biological processes [12,13,14,15]. They are specifically expressed in IBD patients’ peripheral blood and GIT [16,17,18,19].

Among the miRNAs associated with IBD, miR-301a is located on the human chromosome 17q22–17q23 from the miR-130/301 family [20,21,22]. Notably, miR-301a plays an important role in the progression of IBD [20,21,22,23,24]. In addition, miR-301a-knockout mice resist dextran sodium sulfate (DSS)-induced IBD [21,25]. In short, miR-301a could regulate IBD via three mechanisms: (1) miR-301a can facilitate IBD by reducing the tumor necrosis factor α (TNF-α) [26]. (2) miR-301a activates the NF-κB pathway and releases numerous pro-inflammatory factors [21]. (3) miR-301a damages the epithelial integrity and promotes inflammation in the IBD model [25]. Thus, we speculate that inhibiting miR-301a can effectively alleviate IBD via the above mechanisms. Although miR-301a inhibitors (anti-miR-301a) have remarkable therapeutic effects, their degradability and poor ability to target the intestinal inflammation site after oral administration remain unaddressed.

Nanotechnology can bridge the gap between biological and physical sciences as drug delivery systems [26]. Due to its minimal toxicity and tunable rate of biodegradation in vivo, poly (lactic-co-glycolic acid) (PLGA) has been one of the most commonly used and promising polymers in drug delivery systems [27,28]. Moreover, chitosan (CS), an attractive biodegradable and biocompatible polymer, can be combined with PLGA nanoparticles to achieve the multifunction using the cationic primary amino groups of CS [29,30]. CS will increase the interaction duration and enhance the drug penetration [31,32]. Moreover, PLGA and CS have previously been used as gene carriers [33,34,35,36,37]. However, transporting miRNA to the target site and protecting it from degradation are difficult in orally administered nano-based delivery systems [38]. Meanwhile, according to our previous study [39], the electrostatic adsorption of CS was affected at pH 1.2, which poses challenges to the stability and colon-targeting of nanoparticles in miRNA. Eudragit^®^ S100 (ES100), a typical excipient in the pH-sensitive targeting system, has been used to coat the PLGA nanoparticle to achieve a colon targeting ability and slowly release drugs in the GIT [40,41]. Therefore, our group hypothesized that the combination of ES100, PLGA, and CS can effectively deliver anti-miR-301a to the GIT. However, macrophages and epithelial cells have been identified as the essential factors in the progression of intestinal inflammation and tight junctions [42,43]. Therefore, the targeted delivery of drugs to the inflammatory macrophages and epithelial cells is important for the treatment of IBD. CD44 overexpresses on inflammatory macrophages and epithelial cells [44,45,46,47]. Hyaluronan (HA), a non-sulfated glycosaminoglycan, can bind to CD44 [48,49,50]. Our previous study also demonstrated that the cells overexpressing CD44 had an improved cellular uptake of HA-modified nanoparticles [51].

Herein, this paper reports the development of a pH-responsive nanoparticle targeting the CD44 receptor to deliver anti-miR-301a to the colon for IBD treatment. HA-CS/ES100/PLGA nanoparticles (HCeP NP) were prepared using a two-step emulsification and volatility method. The prepared nanoparticles exhibited great stability, pH-responsiveness, colon accumulation, and macrophage targeting ability. Furthermore, pharmacodynamic experiments showed that an oral administration of the anti-miR@HCeP NP to DSS-treated mice improved the disease activity index (DAI), morris score, pro-inflammatory factors levels, and colon length. Moreover, the biocompatibility of the anti-miR-loaded HCeP NP was confirmed. Overall, this nano-drug delivery system is a promising IBD treatment candidate.

## 2. Materials and Methods

### 2.1. Materials and Animals

Anti-miR-301a and FAM-anti-miR were purchased from Shanghai Genepharma Co., Ltd. (Shanghai, China). Poly(lactic-co-glycolic acid) (PLGA, Mw 14kDa) was purchased from Jinan Daigang Biomaterial Co., Ltd. (Jinan, China). Eudragit^®^S100 (ES100) was kindly supplied by Colorcon Co., Ltd. (Shanghai, China). Chitosan (CS, Mw 110 kDa, 90.0% deacetylation degree) and hyaluronic acid (HA, Mw 10 kDa) were purchased from Shanghai Titan Technology Co., Ltd. (Shanghai, China). Polyvinyl alcohol (PVA, Mw 74.8–79.2 kDa) and dichloromethane (DCM) were purchased from Shanghai Lingfeng Chemical Reagent Co., Ltd. (Shanghai, China). 1-ethyl-3-(3-dimethylaminopropyl) carbodiimide (EDC) and N-hydroxysuccinimide (NHS) were purchased from Damas (Shanghai, China). Fluorescein isothiocyanate (FITC) was purchased from Maokang Biotech Co. Ltd. (Shanghai, China). IR-775 was purchased from Sigma-Aldrich (Washington, DC, USA). The enzyme-linked immunosorbent assay (ELISA) kit of TNF-α, lipocalin-2 (LCN-2), Interleukin-6 and 1β (IL-6 and IL-1β) was purchased from Shanghai Enzyme-Linked Biotechnology Co., Ltd. (Shanghai, China). The ELISA kit of myeloperoxidase (MPO) was purchased from Nanjing Jiancheng Bioengineering Institute (Nanjing, China). Lipopolysaccharide (LPS) and pelltobarbitalum natricum (>90%) were purchased from Sigma-Aldrich (Washington, DC, USA). Dextran sulfate sodium (DSS) was purchased from MP Biomedicals (Santa Ana, CA, USA). All other reagents used in this study were analytically pure grade.

Caco-2 and raw 264.7 cells were purchased from the Institute of Biochemistry & Cell Biology, Chinese Academy of Sciences (Shanghai, China). Dulbecco’s modified eagle’s medium (DMEM), Roswell Park Memorial Institute 1640 (RPMI 1640), fetal bovine serum (FBS), phosphate-buffered saline (PBS), trypsin-EDTA and penicillin-streptomycin were all purchased from Gibco (Carlsbad, CA, USA). Hoechst 33342 was purchased from Shanghai Beyotime Biotechnology Co., Ltd. (Shanghai, China).

Balb/c mice (8–12 weeks, 21 ± 1 g) used in the experiments were purchased from Jiesijie Experimental Animal Co., Ltd. (Shanghai, China) and acclimatized to a specific pathogen-free environment under controlled conditions (22 ± 2 °C) for one week. The animal experiments were carried out in accordance with the guidelines evaluated and approved by the ethics committee of School of Pharmacy from East China University of Science and Technology (ethical approval: SHDSYY-2019-2138).

### 2.2. Preparation of Nanoparticles

Anti-miR-301a was dissolved in water, and spermidine was added at a nitrogen over phosphate (N/P) ratio of 15. The mixture was then stirred and rested for 15 min. Gene solution without spermidine was used as a control. Next, 200 μL of the gene solution was added dropwise into 500 μL of the PLGA/dichloromethane solution (20 mg/mL) with magnetic stirring. The mixture was then sonicated at 350 W for 5 min in an ice bath to obtain anti-miR-301a@PLGA nanoparticles (anti-miR@PLGA NPs). Then, 2 mL of 2% PVA/ES100 solution was added to the anti-miR@PLGA NP solution, followed by sonicating at 350 W for 10 min in an ice bath and magnetic stirring at 1000 rpm for 3 h to obtain anti-miR-301a@ES100/PLGA nanoparticles (anti-miR@eP NPs). The CS solution (1 mg/mL, acetic acid solution with pH at 4.5) was added dropwise into the anti-miR@eP NP solution to obtain anti-miR-301a@CS/ES100/PLGA nanoparticles (anti-miR@CeP NPs). HA was activated by reaction with EDC/NHS and added dropwise into the anti-miR@CeP NP solution. After magnetic stirring at room temperature for 4 h, the solution was transferred to a dialysis bag (Mw 8–10 kDa, Shanghai Yuanye Biotechnology Co., Ltd., Shanghai, China) and dialyzed in water (pH 6.8) for 1 h. The nanoparticle solution was lyophilized to obtain the anti-miR-301a@HA-CS/ES100/PLGA nanoparticles (anti-miR@HCeP NPs).

This paper optimized the nanoparticle formulations by varying the PLGA, ES100, CS, and HA concentrations. The FAM-anti-miR@HA-CS/ES100/PLGA nanoparticles (FAM-anti-miR@HCeP NP), IR-775@HA-CS/ES100/PLGA nanoparticles (IR-775@HCeP NP), and FITC@HA-CS/ES100/PLGA nanoparticles (FITC@HCeP NP) used in the subsequent experiments were prepared through the same method using FAM-anti-miR, IR-775, and FITC instead of anti-miR-301a. IR-775 and FITC were dissolved in 2% ethanol solution and 1.5 mg/mL tris solution (pH 7.0), respectively.

### 2.3. Characterization of the Nanoparticles

The mean particle size, polydispersity index (PDI), and zeta potential of nanoparticles were measured using dynamic light scattering (DLS) using a Nano-ZS90 Zetasizer (Malvern, UK). The morphology of the nanoparticles was examined through transmission electron microscopy (TEM). Briefly, the nanoparticle solution was diluted 10 times and vortex to disperse uniformly, 10 μL of diluted solution was dropped onto a clean copper grid and stained with 2% (*w*/*v*) phosphotungstic acid for 20 s, followed by observation using a JEM-2100 instrument (JEOL Ltd., Tokyo, Japan).

The binding ability of nanoparticles to anti-miR-301a was measured via gel electrophoresis experiments. The anti-miR@HCeP NP with different PLGA/anti-miR-301a (*w*/*w*) ratios (0–600) were prepared according to the procedure described above. The nanoparticle solution mixed with loading buffer was applied to a 1% (*w*/*v*) agarose gel in 10% (*v*/*v*) 10× tris borate EDTA (TBE) running buffer containing staining reagent DuRed. The electrophoresis was conducted at 110 V for 45 min. The results were revealed with an ultraviolet-visible (UV) spectrophotometer (GelSMART, DragonLab Co. Ltd., Beijing, China). The encapsulation efficiency (EE) of nanoparticles was analyzed via UV spectrophotometer.

### 2.4. Stability of Nanoparticles

The anti-miR@HCeP NPs were stored at 4 °C in PBS (pH 6.5). Samples were collected daily for one week. The particle size and zeta potential of the nanoparticles were measured using DLS. Furthermore, the FAM-anti-miR@HCeP NPs were collected daily at 4 °C for one week, and centrifuged at 12,000 rpm and 4 °C for 30 min. Next, the supernatant was collected and placed into 96-well plates in the dark. The concentration of the FAM-anti-miR was measured via the multifunctional microplate reader (BioTek, Winooski, VT, USA) for predicting the stability of the anti-miR in nanoparticles. In addition, the stability of nanoparticles was measured via gel electrophoresis experiments. The anti-miR@HCeP NPs, stored at 4 °C, were taken out at 1, 4 and 7 d. Subsequently, the samples were examined using gel electrophoresis as mentioned above. Meanwhile, the anti-miR-301a was used as a control.

### 2.5. Acid Resistance of Nanoparticles

The HCeP NPs were diluted in an HCl solution (pH 1.2) for 2 h, then centrifugated at 8000 rpm for 20 min. Next, the supernatant was removed, and the sediment was dissolved in 0.01 M PBS buffer (pH 6.8) and gently shaken in a 37 °C water bath at 100 rpm for 6 h. As controls, HCeP NPs and eP NPs were dissolved in 0.01 M PBS buffer (pH 6.8) and gently shaken in a 37 °C water bath at 100 rpm for 6 h. The samples were collected at the set time points (0, 2, 4 and 6 h). The acid resistance of nanoparticles was evaluated by measuring particle size and zeta potential in PBS (pH 6.8) as described above.

### 2.6. In Vitro Drug Release

The in vitro drug release from the FAM-anti-miR@HCeP NPs was studied sequentially in different pH buffer solutions (pH 6.8 and 7.4) to simulate the pH environment from the small intestine and the colon. Briefly, the FAM-anti-miR@HCeP NPs were diluted in an HCl solution (pH 1.2) for 2 h (concentration of anti-miR-301a = 5 nM), and the pH was adjusted to 6.8 and 7.4, as well as gently shaken at 100 rpm at 37 °C. At predetermined time intervals (0, 1, 2, 4, 6, 8, 12, 24, and 36 h), 0.2 mL of sample was collected and replaced with fresh release medium. After centrifugation at 12,000 rpm and 4 °C for 30 min, 50 μL of supernatant was collected and placed into 96-well plates in the dark. The fluorescence intensity was measured using a multifunctional microplate reader (BioTek, Winooski, VT, USA).

### 2.7. Cell Culture

Caco-2 cells were cultured in DMEM supplemented with 1% (*v*/*v*) penicillin-streptomycin and 10% (*v*/*v*) FBS at 37 °C in 5% CO_2_ atmosphere with relative humidity of 90%. Raw 264.7 cells were cultured in RPMI1640 medium supplemented with 1% (*v*/*v*) penicillin-streptomycin and 10% (*v*/*v*) FBS at 37 °C in 5% CO_2_ atmosphere with relative humidity of 90%.

### 2.8. Cytotoxicity Assay

The viability of the blank HCeP NP-, the anti-miR@HCeP NP- and FAM-anti-miR@HCeP NP-treated caco-2 and raw 264.7 cells was evaluated through the cell counting kit-8 (CCK-8) method. Briefly, caco-2 and raw 264.7 cells at the logarithmic phase were seeded at a density of 4 × 10^4^ cells per well on 96-well plates and incubated for 24 h. Then, the culture medium was removed and 100 μL of the HCeP NPs (50, 100, 200, or 500 μg/mL) or the anti-miR@HCeP NPs (anti-miR concentration: 0.5, 5, 10, or 30 nM) was added before incubation for 24 h. Before the determination, 10 μL of CCK-8 solution was added into each well and the cells were incubated for 1 h at 37 °C. Then, the absorbance of each well was measured using a microplate reader (BioTek, Winooski, VT, USA) at 450 nm.

### 2.9. Cellular Uptake

Caco-2 cells and raw 264.7 cells at the logarithmic phase were seeded at a density of 1 × 10^5^ cells per well on 24-well plates and incubated for 24 h. Then, the culture medium was replaced with an FBS-free medium containing FAM-anti-miR@HCeP NPs, and the cells were incubated for 4 h at 37 °C. After incubation, the culture medium was removed and the cells were washed twice with cold PBS (pH 6.5). To confirm the cellular uptake mechanism, the cells were cultured with an excess of HA (1 mg/mL) for 1 h and then washed with cold PBS before treatment to competitively inhibit CD44-mediated cellular uptake. Next, the cells were fixed with 10% (*w*/*v*) paraformaldehyde for 20 min and cell nuclei were stained with Hoechst 33342 (10 µg/mL) for 10 min. Then, the cells were washed twice with cold PBS. Finally, cellular uptake was visualized using a fluorescence microscope (Ti-S, Nikon, Tokyo, Japan).

### 2.10. In Vitro Cytokine Assay

Raw 264.7 cells at the logarithmic phase were seeded at a density of 4 × 10^4^ cells per well on 96-well plates and incubated with LPS for 24 h. Then, the culture medium was replaced with the anti-miR@HCeP NP solution (concentration of anti-miR-301a = 20 nM). After incubation for 2 h at 37 °C, the supernatant was collected and TNF-α, IL-6, and IL-1β were quantified using ELISA.

### 2.11. Biodistribution and Colon-Targeting Ability of Nanoparticles

To investigate the biodistribution of nanoparticles, our group randomly assigned the mice to different groups: IBD mice (Balb/C mice induced using 2.5% DSS for 7 d) received free IR-775, IR-775@PLGA NPs, IR-775@eP NPs, or IR-775@HCeP NPs orally and healthy mice (control group) received IR-775@HCeP NPs orally. In addition, other IBD mice were divided into 4 groups and given FITC, FITC@PLGA NPs, FITC@eP NPs, or FITC@HCeP NPs using intragastric administration.

Mice were anesthetized 1, 4, 18 and 36 h after treatment, and real-time fluorescence images were taken using an in vivo imaging system (FX-PRO, Carestream Health, Flowood, MS, USA) at the excitation wavelength of 740 nm and emission wavelength of 790 nm. After 18 h, mice were sacrificed and their gastrointestinal tract was excised. Ex vivo images were obtained as described above. For the immunofluorescence section of colon tissue, after intragastric administration of fluorescent nanoparticles at 18 h, mice were sacrificed, and the colon was removed. Frozen sections of colon tissue were taken and we labeled the nuclei with Hoechst 33342. Finally, the immunofluorescence section was visualized using a fluorescence microscope (Ti-S, Nikon, Tokyo, Japan).

### 2.12. In Vivo Therapeutic Efficacy

The mice were randomly assigned to four groups (five mice in each group): control group, DSS group, free anti-miR-treated group, and the anti-miR@HCeP NP-treated group. Briefly, the DSS group and treated groups were free to drink 2.5% DSS in water from 0 to 7 d, while the control group was free to drink DSS-free sterilized water. The treated groups received gavage of free anti-miR or the anti-miR@HCeP NPs (0.4 mg anti-miR-301a/kg/day) from 4 to 10 d, while the other two groups received gavage of an equal volume of saline [52]. The mice were sacrificed on day 12, and their body weight, colon length, disease activity index (DAI), morris score, histological injury score, spleen-to-body weight ratio, myeloperoxidase (MPO) activity, and LCN-2 and pro-inflammatory factors expression levels were recorded.

Briefly, the DAI score was determined by scoring the body weight loss, diarrheal condition, and fecal bleeding. The morris score was evaluated by observing congestion and colon ulcers. The histological damage was analyzed as follows: the harvested colons of mice were stained with hematoxylin and eosin (H&E). The mucosal structure changes, cell infiltration, inflammation, goblet cell depletion, surface epithelial cell proliferation, and epithelial regeneration were observed under a microscope. The colon mucosa was ground in cold PBS and centrifugated to obtain the supernatant. MPO, TNF-α, IL-6, and IL-1β in colon mucosa were quantified using appropriate ELISA kits according to the manufacturer’s instructions. The feces were ground in cold PBS and centrifugated to obtain the supernatant. LCN-2 in feces was quantified using an LCN-2 ELISA kit according to the manufacturer’s instructions.

### 2.13. Histology

For histological analysis, H&E staining was performed. The paraffin sections of colon tissue were dewaxed and hydrated, and the prepared hematoxylin staining solution was added to the colon tissue sections. After the differentiation was completed, the weakly alkaline blue promoting solution was added to make the hematoxylin appear blue. After this step was complete, eosin dye was added to the tissue section sample and incubated for a few minutes for full color development. After staining, gradient dehydration was carried out, and finally gum was dropped into the sample and sealed with a cover glass. They were photographed and recorded under a microscope. For the immunofluorescence section of colon tissue, after intragastric administration of fluorescent nanoparticles at 18 h, mice were sacrificed, and the colon was removed. We took frozen sections of colon tissue and labeled the nuclei with Hoechst 33342. Finally, the immunofluorescence section was visualized using a fluorescence microscope (Ti-S, Nikon, Tokyo, Japan).

### 2.14. Safety Evaluation

After therapy, whole blood was collected from the orbit of mice before euthanasia, which was then centrifuged to get the serum for the aspartate aminotransferase (AST), alanine aminotransferase (ALT), serum creatinine (CRE), and blood urea nitrogen (BUN) kit-test. The major organs (heart, liver, spleen, lung and kidney) of the mice were collected, fixed, and stained with H&E.

### 2.15. Statistical Analysis

All data were expressed as mean ± standard deviation. Multiple group comparisons were made with one-way analysis of variance (ANOVA). A value of *p* < 0.05 was considered statistically significant.

## 3. Results and Discussion

### 3.1. Characterization of the Nanoparticles

Figure 1A shows the chemical formulas of PLGA, ES100, CS, and HA. Table 1 summarizes the effect of the CS/PLGA and HA/CS weight ratios on the physicochemical properties of the HCeP NPs. Increasing the CS/PLGA weight ratio from 0.1:2 to 0.6:2 decreased the hydrodynamic diameter of the CeP NP from 266.2 ± 11.2 to 212.7 ± 2.4 nm, while sequentially increasing the CS amount induced aggregate precipitation. CS is positively charged and PLGA is negatively charged, thus enabling CS loading through electrostatic interaction. Increasing the amount of CS sufficiently remarkably increased the particle size and PDI, probably due to the enhanced aggregation resulting from the increase in surface charge. This was evidenced by the gradual increase in the zeta potential of the CeP NP from −31.7 ± 0.7 to −3.5 ± 0.2 mV upon an increase in the CS/PLGA weight ratio from 0.1:2 to 0.6:2.

The EDC/NHS catalyst caused a condensation reaction between the carboxyl groups of HA and the amine groups of CS. Meanwhile, the amount of HA considerably affected the particle size and zeta potential of HCeP NPs. Increasing the HA/CS weight ratio from 0.2:1 to 0.5:1 increased the HCeP NP particle size from 237.2 ± 1.6 to 261.1 ± 6.3 nm, and decreased the zeta potential from −15.6 ± 0.4 to −20 ± 0.2 mV. However, a high HA amount increased the zeta potential of HCeP NPs, which reached −13.8 ± 1.3 mV when the HA/CS weight ratio was increased to 2:1. In the rest of the study, we used CS/PLGA and HA/CS weight ratios of 0.4:2 and 0.5:1, respectively. The resulting HCeP NPs had a hydrodynamic diameter of 261.1 ± 6.3 nm, a zeta potential of −20 ± 0.2 mV, and a PDI of 0.26.

The mean particle size of the anti-miR@HCeP NPs was 272.5 ± 1.7 nm, the zeta potential was −24.6 ± 0.1 mV, and the PDI was 0.22 ± 0.03. Figure 1B shows the possible formulation mechanism of the anti-miR@HCeP NPs. The refined anti-miR@HCeP NPs also exhibited an acceptable intensity distribution (Figure 1C) and uniform particle size with a spheroid shape, as confirmed through TEM (Figure 1D).

This paper assessed the anti-miR-301a encapsulation efficiency of HCeP NPs using agarose gel electrophoresis. As shown in Figure 1E, adding spermidine allowed the nanoparticles to retain the anti-miRNA. This might be because the anti-miR-301a and the surface of HCeP NPs are both negatively charged. As the equilibrium ion of gene aggregation, spermidine might cause the charge inversion of genes during the coagulation process, which helped the anion polymer to retain them. Thus, spermidine is necessary to load the genes into the nanoparticles, and the PLGA/anti-miR-301a weight ratio of 300:1 yielded the highest entrapment efficiency. Furthermore, the encapsulation efficiency of anti-miR@HCeP NPs was 73.5%. The above result indicated that the anti-miR@HCeP NPs prepared in this paper had great physicochemical properties, as well as gene loading advantages.

### 3.2. Storage Stability

Figure 1F shows that the particle size and zeta potential of the anti-miR@HCeP NPs only marginally changed over 7 d. This result indicated that the anti-miR@HCeP NPs had a good storage stability. Furthermore, as shown in Figure 1F, the concentration of the FAM-anti-miR exhibited only a slight variation during the week, indicating the superior stability of loading anti-miR. Meanwhile, the gel electrophoresis result of the anti-miR@HCeP NPs showed that nanoparticles could compress the gene effectively at 1, 4, and 7 d (Appendix A). The above results further indicate the superior stability of anti-miR@HCeP NPs.

### 3.3. Acid Resistance

To investigate whether gastric acid would remove the CS shell from HCeP NPs, HCeP NPs were dissolved in an HCl solution (pH 1.2) for 2 h and then we transferred the mixture into 0.01 M PBS buffer (pH 6.8) for 6 h. Next, the acid resistance of nanoparticles was evaluated by measuring their size and zeta potential in PBS (pH 6.8). Figure 1G shows that the particle size and zeta potential of HCl-treated HCeP NPs were comparable to those of untreated HCeP NPs. In addition, the zeta potential of eP NPs was between −34.7 and −33.1 mV, while that of HCl-treated HCeP NPs changed from −22.6 mV to −21.4 mV, indicating that the nanoparticles conserved their CS shell. Thus, HCeP NPs could potentially resist gastric acid and reach the colon. However, the in vitro drug release of HCeP NPs with anti-miR is also needed.

### 3.4. In Vitro Drug Release

In vitro, the release of FAM-anti-miR@HCeP NPs was unnoticed after being dissolved in the HCl solution (pH 1.2) for 2 h. Furthermore, the FAM-anti-miR@HCeP NPs (Figure 1H) released 7.2% of the anti-miR-301a at pH 6.8 and 36.2% at pH 7.4 within the first 2 h. The cumulative release rates of anti-miR-301a at 36 h were 19.6% at pH 6.8, and 70.3% at pH 7.4. The above results indicated that the ES100 of anti-miR@HCeP NPs dissolved in an alkaline environment. These results also demonstrated the pH sensitivity of anti-miR@HCeP NPs, and indicated that the drug system could be retain in the small intestine, as well as released in the colon. This result also indicated that anti-miR@HCeP NPs have a stronger application prospect in gastrointestinal diseases than CS and PLGA nanoparticles.

### 3.5. Cytotoxicity Assay

During intestinal inflammation, macrophages produce pro-inflammatory cytokines that exacerbate the inflammatory response [53]. In addition, the disruption of intestinal barriers and tight junctions can aggravate inflammatory symptoms [43]. Caco-2 cells are a human clone of colon adenocarcinoma cells that are structurally, functionally, and morphologically similar to human small intestinal epithelial cells with the same tight junctions [54]. Therefore, the modulation of macrophage and caco-2 cells is crucial for IBD. Before confirming the effect, the safety of nanocarriers to raw 264.7 and caco-2 cells needed to be evaluated.

A CCK-8 assay revealed no cytotoxicity in blank HCeP NP- and anti-miR@HCeP NP-treated raw 264.7 and caco-2 cells after 24 h (Figure 2A). The cell viability of HCeP NPs remained acceptable even at 500 μg/mL. This result indicated made our research group believe that the anti-miR@HCeP NPs are safe. The nanoparticle concentrations used in the subsequent experiments were below 500 μg/mL. Furthermore, the safety of FAM-anti-miR@HCeP NPs was confirmed in Appendix A. Therefore, our subsequent cytological investigations were carried out at safe concentrations.

### 3.6. Cellular Uptake

Next, this paper evaluated the ability of caco-2 and raw 264.7 cells to uptake FAM-anti-miR@HCeP NPs. Figure 2B shows that both cell lines internalized the HCeP NPs.

To confirm the role of HA in cellular uptake, raw 264.7 cells were incubated with excess HA before the cellular uptake experiment. As shown in Figure 2B, HA-treated raw 264.7 cells exhibited lower green fluorescence signals than untreated raw 264.7 cells. This was because CD44 is expressed on the surface of macrophages in intestinal inflammatory tissues [55]. CD44 saturation inhibited HA-promoted cellular uptake. Therefore, by modifying HA on the surface of nanoparticles, the uptake capacity of macrophages could be improved obviously. The above results showed the excellent cell targeting of FAM-anti-miR@HCeP NPs.

### 3.7. In Vitro Cytokine Assay

LPS-stimulated raw 264.7 cells secreted inflammatory cytokines such as TNF-α, IL-1β, and IL-6. Figure 2C showed that the anti-miR@HCeP NP treatment drastically reduced the secretion of inflammatory cytokines. The activation of the NF-κB pathway in the LPS-stimulated raw 264.7 cells induced the immune response of the macrophages, followed by the secretion of inflammatory cytokines. Huang demonstrated that the NF-κB-repressing factor (NKRF) is a downstream target gene of miR-301a, which downregulates NF-κB [56]. Thus, the miR-301a inhibitor increased the NKRF expression in macrophages, inhibiting the NF-κB pathway and reducing inflammatory cytokines secretion. This result suggested that NF-κB can be a potential target of anti-miR@HCeP NPs in IBD development.

### 3.8. Biodistribution and Colon-Targeting Ability of Nanoparticles

To reveal the in vivo distribution of the HCeP NPs, IR-775 chloride, a near-infrared fluorescent dye with an excitation wavelength of 774 nm and an emission wavelength of 792 nm, was adopted. Briefly, mice received free IR-775, IR-775@PLGA NPs, IR-775@eP NPs, or IR-775@HCeP NPs orally and we measured fluorescence after 0, 1, 4, 18, and 36 h. As shown in Figure 3A, the fluorescence of the free IR-775 group decreased gradually over time and was almost undetectable at 36 h. Thus, IR-775 was metabolized out of the body within 36 h. However, our group observed higher fluorescence signals than in the free IR-775 group after 18 h, and some fluorescence remained at 36 h in the nanoparticle groups. This might be attributed to nanoparticles protecting the fluorescent substance from metabolism and the biological adhesion of polymer nanoparticles extending the retention time of drugs in the gastrointestinal tract. These results were consistent with results of in vitro release studies.

Next, the effect of nanoparticles on inflammation was investigated. DSS primarily inflames the rectum and colon, and might damage the small intestine. As shown in Figure 3B, HCeP NPs were mainly concentrated in the intestinal tract of IBD mice, especially in the colon. However, there was no significant fluorescence in the colon of healthy mice, which indicated the targeting ability of drug delivery system at the inflammation site. The possible explanations for this are that the barrier at the inflammatory site of the intestinal tract was disrupted, the tight junctions of intestinal cells were weakened, or the permeability was increased [57]. Furthermore, the infiltration of the immune cells may have resulted in the accumulation of nanoparticles at the intestinal inflammatory site [58].

Moreover, PLGA NPs, eP NPs, and HCeP NPs were mainly concentrated in the intestinal tract while free IR-775 was not. This suggests that nanoparticles can increase the concentration of drugs in the intestine. The IR775@PLGA NP group exhibited stronger fluorescence intensity than the IR-775@eP NP group in the stomach and proximal small intestine, suggesting that PLGA NPs were mainly concentrated in the stomach and proximal small intestine. The eP NPs coated with an enteric material concentrated more in the distal small intestine and colon than in the stomach and proximal small intestine. Moreover, the HCeP NPs exhibited better colonic targeting ability than the eP NPs, probably thanks to the HA on the surface of the HCeP NPs.

Figure 3C shows colonic tissue cryo-sections from mice with DSS-induced colitis treated with free FITC, FITC@PLGA NPs, FITC@eP NPs, and FITC@HCeP NPs. There was almost no fluorescence accumulation in the colon of the free FITC-treated mice, possibly due to the rapid metabolism of free FITC and the intestinal cells’ poor uptake ability of free drugs. FITC@PLGA NPs were more concentrated around the colonic epithelial cells than free FITC, but only a small amount of FITC entered the tissue. The FITC@eP NPs were more concentrated in colon tissue than the FITC@PLGA NPs, probably because of the enteric coat. The FITC@HCeP NPs were more concentrated in the colonic inflammatory site and entered the intestinal tissue more deeply than the FITC@eP NPs. Figure 3D shows that the quantification results were consistent with the above results. Therefore, the colon targeting properties of HCeP NPs were also trustworthy.

### 3.9. In Vivo Therapeutic Efficacy

Next, the in vivo therapeutic efficacy of anti-miR@HCeP NPs was evaluated. Figure 4A shows a schematic diagram of the anti-miR@HCeP NPs administration and IBD induction course applied to the mice. Then, this paper monitored the body weight, colon length, DAI, morris score, histological injury score, spleen-to-body weight ratio, MPO activity, and LCN-2 and pro-inflammatory factors expression. The results showed that anti-miR@HCeP NPs had great therapeutic effects on IBD. Compared with anti-miR, anti-miR@HCeP NPs have significant therapeutic advantages due to the colon-targeting and inflammation-targeting properties of HCeP NPs.

### 3.10. Bodyweight and DAI

Figure 4B,C show the body weight and DAI of the different mouse groups over time, respectively. The body weight of IBD mice began to decline from 4 d, and their DAI score increased from 0 d. The stool morphology changed from granular to sparse, and a bloody stool was observed. Later, the mice started to curl up, stopped eating and drinking, and some even died. Compared with the DSS group, there was no significant weight increase or remission of disease activity in the free anti-miR group. The mice in the anti-miR@HCeP NP group initially gained weight, although not as quickly as those in the control group did, and then lost body weight, albeit not as fast as those in the DSS group. The stool of anti-miR@HCeP NP-treated mice gradually retrieved a normal aspect. Moreover, their DAI began to decrease on 5 d and reached 2 on 12 d. The result further indicated the notable therapeutic effect of anti-miR@HCeP NPs in IBD mice.

### 3.11. Morris Score and Colon Length

Table 2 shows that the morris score of the DSS group was 6.8 ± 0.4, that of the free anti-miR group was 6.5 ± 0.3, while that of the anti-miR@HCeP NP group decreased to 1.3 ± 0.2. Figure 4D,E show that the DSS group mice had a drastically shortened colon. Healthy and DSS group mice had average colon lengths of approximately 8.5 and 5.5 cm, respectively. Moreover, the free anti-miR group had a colon length similar to the DSS group. Meanwhile, anti-miR@HCeP NP-treated mice had a colon length similar to the control group mice. Our group observed that DSS group mice had a narrow intestinal cavity, weak elasticity, and hyperemic ulcerations in the gut. However, mice in the control and anti-miR@HCeP NP groups had healthy colons with great elasticity and without hyperemic ulcers. The above results further verified the therapeutic effect of anti-miR@HCeP NPs.

### 3.12. Histological Injury Assessment

Figure 4F shows that the anti-miR@HCeP NP group had a significantly lower histopathological score than the DSS group after therapy. However, we assessed histological injury by performing H&E staining on colon sections. As shown in Figure 4G, the control group exhibited a healthy colon tissue morphology and no inflammation. Meanwhile, the colon tissue of the DSS group showed extensive colonic mucosa disappearance, goblet cell reduction, and colonic crypt destruction. The disordered cell arrangement and extensive infiltration of inflammatory cells were also observed. Free anti-miR did not noticeably alleviate the histological injuries. The anti-miR@HCeP NP group only displayed slight colonic crypt damage and inflammatory cell infiltration. These results also indicated the therapeutic effect of anti-miR@HCeP NPs, which was consistent with the above in vivo therapeutic efficacy result.

### 3.13. Spleen Mass to Body Weight Ratio

As the spleen is an important immune organ, it is enlarged by a series of immune responses when inflammation occurs. As shown in Figure 4H, the spleen-to-body weight ratio in healthy mice stayed at approximately 3 but was significantly higher in the DSS group. Meanwhile, the spleen-to-body weight ratio of anti-miR@HCeP NPs was significantly decreased. These results also suggested that anti-miR@HCeP NPs have a great therapeutic effect.

### 3.14. Evaluation of MPO and LCN-2 Activity

Myeloperoxidase (MPO), produced via neutrophils, could induce oxidative damage to tissues and promote the inflammation of IBD [59]. Lipocalin-2 (LCN-2) is overexpressed in colonic epithelium during IBD [11]. Therefore, MPO and LCN-2 activity in the colon were assessed as additional parameters of disease activity. In Figure 4I, the intestinal mucosa of healthy mice contained approximately 0.27 U/g of MPO. The DSS group mice had 1.08 U/g of MPO, which was four times more than healthy mice. In the anti-miR@HCeP NP group, the MPO level was 58% lower than in the DSS group. This result indicated that anti-miR@HCeP NPs relieved inflammation. As shown in Figure 4J, the feces of healthy mice contained approximately 642 ng/g of LCN-2. The DSS group’s feces contained more LCN-2 than the control group, indicating inflammation. In the anti-miR@HCeP NP group, the LCN-2 level was 37.5% lower than in the DSS group, indicating the anti-inflammatory potential of anti-miR@HCeP NPs.

### 3.15. Pro-Inflammatory Cytokines TNF-α, IL-6, and IL-1β Quantification

TNF-α, IL-6, and IL-1β levels were low in healthy mice and high in DSS group mice. The anti-miR@HCeP NP group had TNF-α, IL-6, and IL-1β levels 2.0, 2.6, and 2.1 times lower than the DSS group, respectively (Figure 4K–M). The pro-inflammatory cytokines levels of the anti-miR@HCeP NP and control groups were similar. These results indicated that the anti-miR@HCeP NPs significantly reduced the levels of inflammatory cytokines in IBD.

### 3.16. Safety Evaluation

The major concern of nano-drug delivery systems is an unexpectable harmfulness to normal tissues. Hence, we examined the major organs (heart, liver, spleen, lung, kidney) via H&E staining. Obvious pathologic abnormalities were not observed in any group after therapy (Figure 5A). In addition, the liver and the kidney function were related to the clinical biochemical degree, such as aspartate aminotransferase (AST), alanine aminotransferase (ALT) and creatinine (CRE), as well as blood urea nitrogen (BUN) [11]. The values of AST, ALT, BUN, and CRE were in the normal range compared with the healthy group (Figure 5B). All the above data suggested that anti-miR@HCeP NPs would not cause toxicity at the systemic level. These results also ensured the safety advantage of anti-miR@HCeP NPs, with their anti-inflammatory ability, in the treatment of IBD.

## 4. Conclusions

Gene therapy is a current trend in drug research, but the rapid in vivo degradation and elimination of genetic material have limited its development. This study prepared the anti-miR@HCeP NPs using a two-step emulsification and volatility method and electrostatic absorption, followed with an EDC/NHS reaction. Furthermore, the acid resistance mechanism of the anti-miR@HCeP NPs using media with varying pH to imitate the oral administration process was confirmed, and showed that nanoparticles provided drug release features. It also proved that nanoparticles with a colon-accumulating ability and pH-responsive for IBD treatment were developed. In addition, our group demonstrated that the anti-miR@HCeP NPs possessed the innocuity and cell-penetrating ability of macrophages and Caco-2 cells in vitro. DSS-induced IBD mouse models were used to show the excellent therapeutic effects on the DAI, morris score, colon length, levels of several pro-inflammatory factors, and histopathology via the anti-miR@HCeP NPs. Importantly, the safety of the anti-miR@HCeP NPs was also confirmed in this study. These details indicated that the anti-miR@HCeP NPs effectively alleviated the inflammatory response and the broken intestinal barrier, proposing a promising strategy for the clinical application of the anti-miR in IBD. Meanwhile, this strategy expands the clinical potential of CS and PLGA nanodelivery systems for oral gene delivery. Therefore, anti-miR@HCeP NPs can be used as a promising nanotherapeutic for IBD treatment via the oral route. However, the complex carrier composition may limit the industrialization and application of anti-miR@HCeP NPs. Fortunately, with the development of microfluidic technology, the industrialization of complex carriers has been gradually overcome. Additionally, the emergence of multifunctional materials also provides a certain idea for simplifying the preparation of complex nanoparticles. Overall, on the basis of providing new ideas for the treatment of IBD patients, the anti-miR@HCeP NPs also provide a new option for the oral delivery of gene drugs, which has a great prospect along with the upgrading of existing technologies.

## Figures and Tables

**Figure 1 nanomaterials-13-02797-f001:**
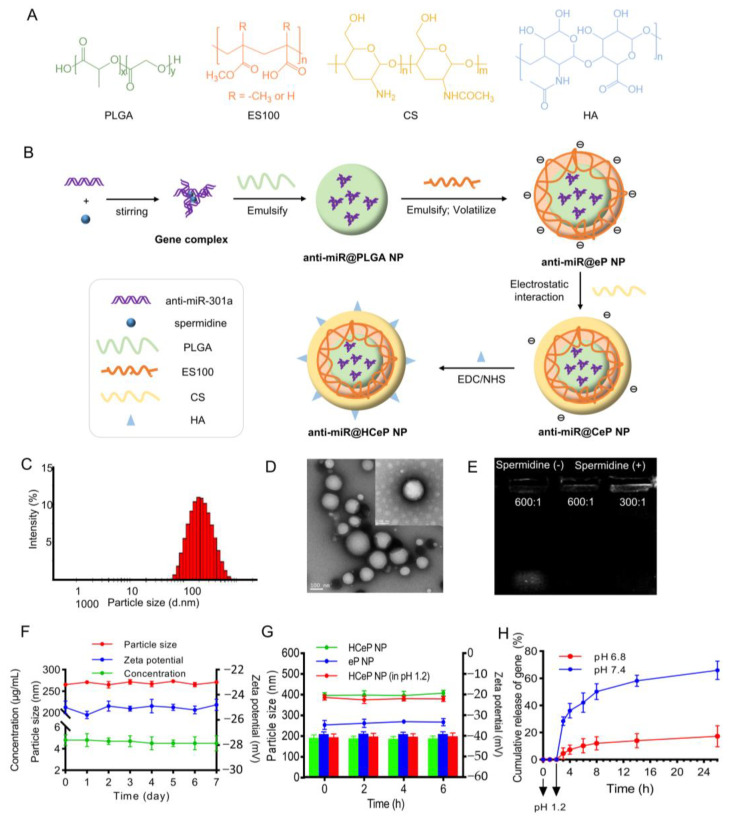
Characterization of the anti-miR@HCeP NP. (**A**) The chemical formula of the nanoparticle polymers. (**B**) The possible formulation mechanism of the gene-loaded nanoparticles. (**C**) The diameter distribution and (**D**) TEM images of anti-miR@HCeP NPs. (**E**) Electrophoretogram of anti-miR@HCeP NPs with and without spermidine. PLGA/anti-miR-301a weight ratio: 600:1; 300:1. (**F**) The stability of anti-miR@HCeP NPs or FAM-anti-miR@HCeP NPs over 7 d. (**G**) The particle size and zeta potential of nanoparticles change at different time points under different conditions. The columns represent particle size, and the lines represent zeta potential. (**H**) In vitro accumulative release profiles of anti-miR@HCeP NPs. Data are presented as mean ± SD (n = 3).

**Figure 2 nanomaterials-13-02797-f002:**
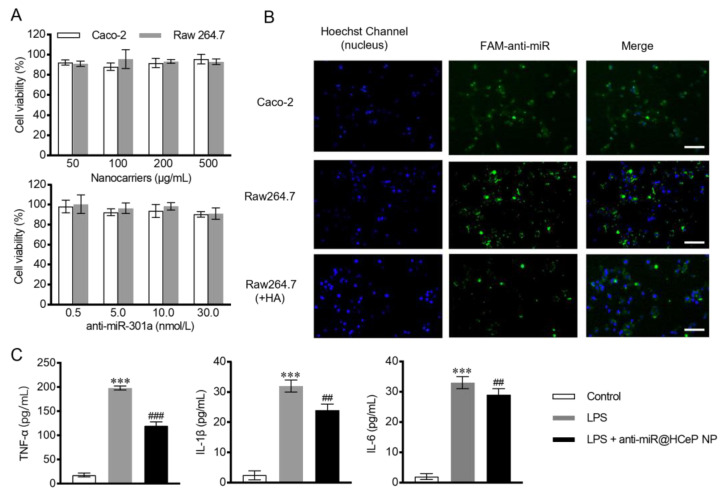
In vitro cytotoxicity, cellular uptake, and inflammatory cytokines levels experiments. (**A**) Cytotoxicity of HCeP NPs and anti-miR@HCeP NPs on caco-2 cells and raw 264.7 cells. (**B**) In vitro cellular uptake of the nanoparticles by caco-2 and raw 264.7 cells (Scale bar: 200 μm). Raw 264.7 cells were incubated with 1 mg/mL of HA for 1 h before the uptake experiment. (**C**) The effect of anti-miR@HCeP NPs (anti-miR-301a = 20 nM) on TNF-α, IL-1β, and IL-6 secretion with raw 264.7 cells. Cells were treated with LPS (0.5 μg/mL) for 24 h for inflammatory cytokines secretion. *** *p* < 0.001 compared with the control group, ^##^
*p* < 0.01, ^###^
*p* < 0.001 compared with the LPS group. Data are presented as mean ± SD (n = 5).

**Figure 3 nanomaterials-13-02797-f003:**
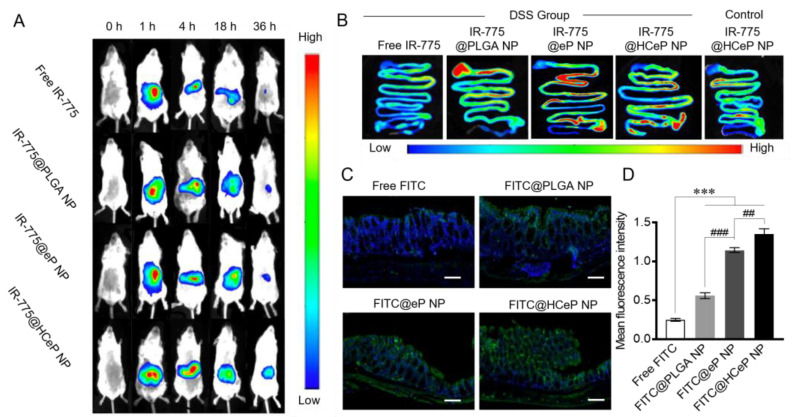
In vivo biodistribution of nanoparticle HCeP. (**A**) In vivo fluorescence images of IBD mice 0, 1, 4, 18, and 36 h after oral administration of free IR-775, IR-775@PLGA NPs, IR-775@eP NPs, and IR-775@HCeP NPs (0.5 mg of IR-775/kg). (**B**) Gastrointestinal fluorescence distribution 18 h after oral administration of free IR-775, IR-775@PLGA NPs, IR-775@eP NPs, and IR-775@HCeP NPs (0.5 mg of IR-775/kg). (**C**) Cryo-sections of colon tissue from IBD mice (scale bar: 50 μm) and (**D**) quantitative analysis 18 h after oral administration of free FITC, FITC@PLGA NPs, FITC@eP NPs, and FITC@HCeP NPs (0.5 mg of FITC/kg). Original magnification: ×5. *** *p* < 0.001 compared with the Free FITC group, ^##^
*p* < 0.01, ^###^
*p* < 0.001 compared with the FITC@eP NPs group.

**Figure 4 nanomaterials-13-02797-f004:**
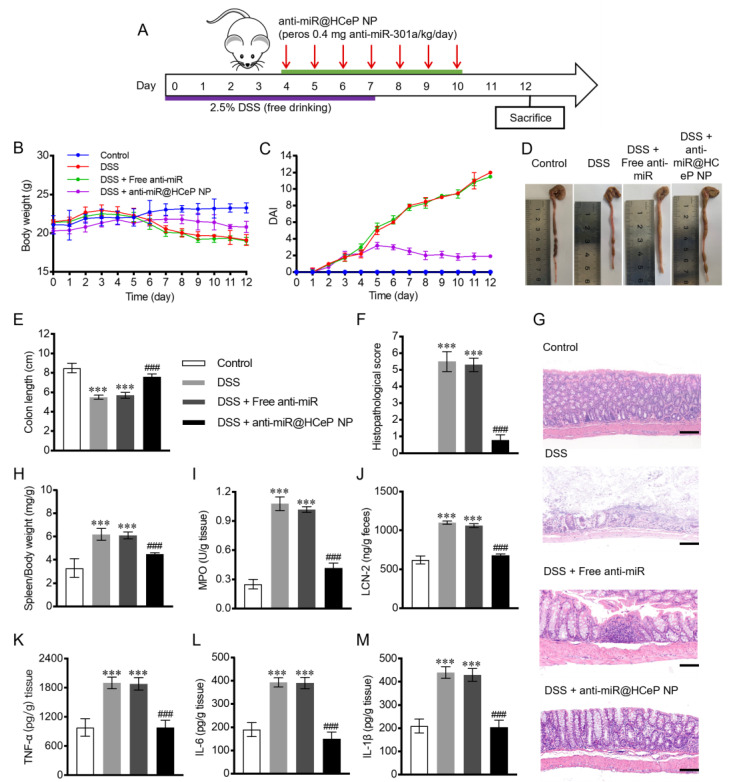
Evaluation of the therapeutic effect of anti-miR@HCeP NPs. (**A**) Schematic diagram of the anti-miR@HCeP NPs administration and IBD induction course. (**B**) Bodyweight and (**C**) DAI. (**D**) Representative photograph of the colons for each group. (**E**) Variations in colon length and (**F**) histopathological score of colon tissue. Original magnification: ×5. (**G**) Representative H&E-stained colon sections for the experimental groups (scale bar: 50 μm) and (**H**) spleen/body weight. (**I**) Colonic MPO activity and (**J**) fecal LCN-2 concentration. Inflammatory cytokines (**K**) TNF-α, (**L**) IL-6, (**M**) IL-1β levels in the colon tissue of anti-miR@HCeP NP-treated mice. *** *p* < 0.001 compared with the control group, ^###^
*p* < 0.001 compared with the DSS group. Data are presented as mean ± SD (n = 5).

**Figure 5 nanomaterials-13-02797-f005:**
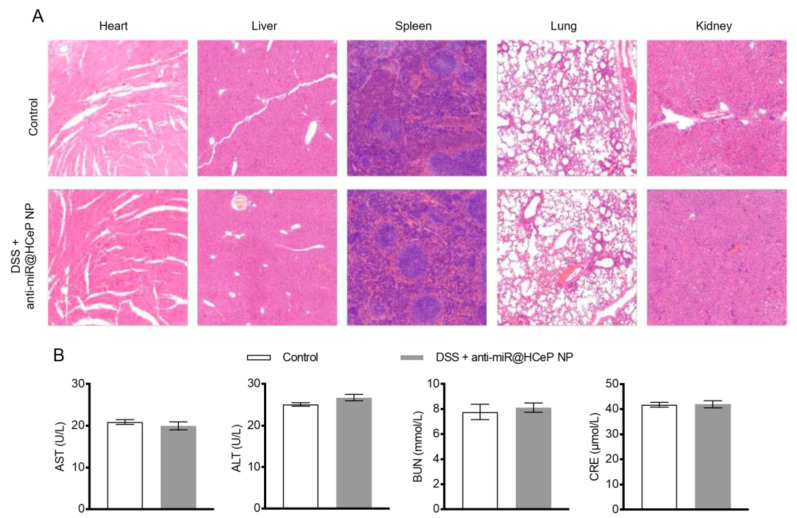
Safety evaluation of anti-miR@HCeP NPs. (**A**) Hematoxylin and eosin-stained dissected organs. Original magnification: ×2. Liver and kidney function markers in the serum of different treated groups. Levels of (**B**) AST, ALT, BUN, and serum CRE in each treatment group. Data were presented as means ± SD (n = 5).

**Table 1 nanomaterials-13-02797-t001:** Physicochemical property of blank HCeP NPs.

Nanocarrier	PLGA:ES:CS:HA (mg/mg/mg/mg)	Particle Size (nm)	Zeta Potential (mV)	PDI
PLGA NP	1/0/0/0	237.5 ± 1.4	−6.1 ± 0.2	0.22
eP NP	2/1/0/0	272.5 ± 1.6	−34.6 ± 0.1	0.22
CeP NP	2/1/0.1/0	266.2 ± 11.2	−31.7 ± 0.7	0.33
	2/1/0.2/0	256.1 ± 5.4	−23.9 ± 0.7	0.28
	2/1/0.4/0	232.9 ± 2.7	−10.9 ± 0.3	0.27
	2/1/0.6/0	212.7 ± 2.4	−3.5 ± 0.2	0.27
	2/1/1/0	__^a^	N/A	N/A
HCeP NP	2/1/0.4/0.08	237.2 ± 1.6	−15.6 ± 0.4	0.24
	2/1/0.4/0.2	261.1 ± 6.3	−20.0 ± 0.2	0.26
	2/1/0.4/0.4	267.5 ± 8.9	−17.5 ± 0.5	0.35
	2/1/0.4/0.8	268.3 ± 6.8	−13.8 ± 1.3	0.39

Notes: ^a^ Precipitation of aggregates was observed. Data were presented as mean ± SD (n = 3). Abbreviations: N/A, not available; PDI, polydispersity; eP NP, Eudragit S100/PLGA nanoparticles; CeP NP, chitosan/Eudragit S100/PLGA nanoparticles; HCeP NP, hyaluronic acid-chitosan/Eudragit S100/PLGA nanoparticles.

**Table 2 nanomaterials-13-02797-t002:** Morris score of different group.

Group	Morris Score
Healthy	0
DSS	6.8 ± 0.4
DSS + Free anti-miR	6.5 ± 0.3
DSS + anti-miR@HCeP NPs	1.3 ± 0.2

Notes: Data were presented as mean ± SD (n = 5). Abbreviations: DSS, dextran sulfate sodium; anti-miR, anti-miR-301a; anti-miR@HCeP NPs, anti-miR-301a@hyaluronic acid-chitosan/Eudragit S100/PLGA nanoparticles.

## Data Availability

Not applicable.

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
