# Peer review of "pH-Sensitive Nanoparticles for Colonic Delivery Anti-miR-301a in Mouse Models of Inflammatory Bowel Diseases"

_nanomaterials, 2023, doi:10.3390/nano13202797_

Round 1

Reviewer 1 Report

Gene therapy is a current trend in drug research. However the rapid in vivo degradation and elimination of genetic material have limited its development. For efficient gene transfection, multiple steps are required including DNA complexation, cellular uptake of the complexes, release of DNA or complexes from endosomes, release of DNA from the carriers, and DNA transfer into the nucleus [30–32]. Inefficient release of the DNA/polymer complex from endocytic vesicles into the cytoplasm is one of the primary causes of poor gene delivery.

The presented study aimed to develop the pH-sensitive nanoparticles based on poly (lactic-co-glycolic acid) copolymer (PLGA), Eudragit® S100 (ES100), chitosan (CS), and hyaluronic acid (HA) as peroral drug delivery system for target delivery of  Anti-miRNA (anti-miR-301a) got the efficient treatment of the  inflammatory bowel diseases (IBD) .

The study is large-scale, multifaceted and shows the effectiveness in vivo and in vitro systems. Interesting aspect of the study and important for the successful treatment of IBD is targeting delivery of drugs to the inflammatory macrophages and epithelial cells.  This technology is based on the overexpressing of CD44 receptor on these cells as the essential factor in the progression of the intestinal inflammation. Hyaluronan - HA-modified nanoparticles, containing a non-sulfated glycosaminoglycan which bind to CD44 receptor is used as specific ligand to macrophages and epithelial cells show an improved cellular uptake of the drug.

However, there are also comments:

1) According to the formulation of the problem, the disadvantage of the system is seen in the fact that a too multicomponent system containing too many components has been created so that it hardly can be implemented as a drug formulation for real use in therapy.

2) poly (lactic-co-glycolic acid) – containing system, biodegradation of which is accompanied by the release of lactic and glycolic acid. As is known from the literature, it is not recommended for the delivery of genetic material because it destabilizing effect on the RNA (even if in combination with chitosan).

3) The need for the ES100 component in the system is not clear enough.

4) The data are not sufficiently discussed in terms of comparison with the systems proposed in the literature.

5) The conclusions largely repeat the abstract and do not contain the nature of the discussion of problems and prospects of the advantages and disadvantages of the developed system and its place among other systems.

Minor  editing of English language required

Author Response

Review 1

1) According to the formulation of the problem, the disadvantage of the system is seen in the fact that a too multicomponent system containing too many components has been created so that it hardly can be implemented as a drug formulation for real use in therapy.

References

  1. Herrera, V.L.; Colby, A.H.; Ruiz-Opazo, N.; Coleman, D.G.; Grinstaff, M.W. Nucleic acid nanomedicines in Phase II/III clinical trials: translation of nucleic acid therapies for reprogramming cells. Nanomedicine (Lond) 2018, 13, 2083–2098.

2) poly (lactic-co-glycolic acid) – containing system, biodegradation of which is accompanied by the release of lactic and glycolic acid. As is known from the literature, it is not recommended for the delivery of genetic material because it destabilizing effect on the RNA (even if in combination with chitosan).

Answer: Really appreciate for your professional and pertinent comments. PLGA can be used in a wide variety of materials, in combination with other materials, to control the amount, while efficiently achieving gene loading and delivery [1-3]. We significantly reduced the amount of PLGA by using spermidine (Figure 1 E). Therefore, we believe that the system in this paper can effectively overcome your concerns.

References

  1. Chen, M.; Gao, S.; Dong, M.; Song, J.; Yang, C.; Howard, K.A.; Kjems, J.; Besenbacher, F. Chitosan/siRNA nanoparticles encapsulated in PLGA nanofibers for siRNA delivery. ACS Nano 2012, 6, 4835–4844.
  2. Miele, D.; Xia, X.; Catenacci, L.; Sorrenti, M.; Rossi, S.; Sandri, G.; Ferrari, F.; Rossi, J.J.; Bonferoni, M.C. Chitosan oleate coated PLGA nanoparticles as siRNA drug delivery system. Pharmaceutics 2021, 13, 1716.
  3. He, Y.; Chen, Q.-W.; Yu, J.-X.; Qin, S.-Y.; Liu, W.-L.; Ma, Y.-H.; Chen, X.-S.; Zhang, A.-Q.; Zhang, X.-Z.; Cheng, Y.-J. Yeast cell membrane-camouflaged PLGA nanoparticle platform for enhanced cancer therapy. J Control Release 2023, 359, 347–358.

3) The need for the ES100 component in the system is not clear enough.

Answer: Really appreciate for your professional and pertinent comments. Transporting miRNA to the colon and protecting it from degradation is difficult in orally administered nano-based delivery systems. Eudragit® S100 (ES100), a typical excipient in pH-sensitive targeting system, has been used to coated PLGA nanoparticle to achieve colon target ability [1]. Meanwhile, according to our previous study [2], the electrostatic adsorption of CS was affected at pH 1.2, which poses challenges to the stability and colon-targeting of nanoparticles in miRNA. Therefore, modification of ES100 on the surface of nanoparticles can achieve colon-targeting properties and acid resistance by pH sensitivity of ES100. Relevant information has been added and marked in the manuscripts.

References

  1. El-Maghawry, E.; Tadros, M.I.; Elkheshen, S.A.; Abd-Elbary, A. Eudragit®-S100 Coated PLGA Nanoparticles for Colon Targeting of Etoricoxib: Optimization and Pharmacokinetic Assessments in Healthy Human Volunteers. Int J Nanomedicine 2020, 15, 3965–3980.
  2. Lv, Y.; Ren, M.; Yao, M.; Zou, J.; Fang, S.; Wang, Y.; Lan, M.; Zhao, Y.; Gao, F. Colon-specific delivery of methotrexate using hyaluronic acid modified pH-responsive nanocarrier for the therapy of colitis in mice. Int J Pharm 2023, 635, 122741.

4) The data are not sufficiently discussed in terms of comparison with the systems proposed in the literature.

Answer: We gratefully thank the reviewer for the great advice. We have supplemented the results and conclusions based on your suggestions.

5) The conclusions largely repeat the abstract and do not contain the nature of the discussion of problems and prospects of the advantages and disadvantages of the developed system and its place among other systems.

Answer: We gratefully thank the reviewer for the great advice. We have an in-depth discussion on the conclusions section based on your suggestions. Gene therapy is a current trend in drug research, but the rapid in vivo degradation and elimination of genetic material have limited its development. This study prepared the anti-miR@HCeP NP using a two-step emulsification and volatility method and electrostatic absorption, followed via EDC/NHS reaction. Furthermore, the acid resistance mechanism of the anti-miR@HCeP NP using media with varying pH to imitate the oral administration process was confirmed, and showed that nanoparticles provided drug release features. It also proved that nanoparticles with colon-accumulating ability and pH-responsive for IBD treatment were developed. In addition, our group demonstrated that the anti-miR@HCeP NP possessed the innocuity and cell-penetrating ability of macrophages and Caco-2 cells in vitro. DSS-induced IBD mouse models were used to show excellent therapeutic effects on DAI, Morris score, colon length, levels of several pro-inflammatory factors, and histopathology via the anti-miR@HCeP NP. Importantly, the safety of the anti-miR@HCeP NP was also confirmed in this study. These details indicated that the anti-miR@HCeP NP effectively alleviated the inflammatory response and the broken intestinal barrier, proposing a promising strategy for the clinical application of the anti-miR in IBD. Meanwhlie, this strategy expands the clinical potential of CS and PLGA nanodelivery systems for oral gene delivery. Therefore, the anti-miR@HCeP NP can be a promising nanotherapeutic for IBD treatment via the oral route. However, the complex carrier composition may limit the industrialization and application of anti-miR@HCeP NP. Fortunately, with the development of microfluidic technology, the industrialization of complex carriers has been gradually overcome. Additionally, the emergence of multifunctional materials also provides a certain idea for simplifying the preparation of complex nanoparticles. Overall, On the basis of providing new ideas for the treatment of IBD patients, the anti-miR@HCeP NP also provides a new option for oral delivery of gene drugs, which has a great prospect along with the upgrading of existing technologies.

Reviewer 2 Report

In this study, the authors synthesized polymeric nanoparticles (NPs) to deliver an anti-inflammatory gene inhibitors for bowel syndrome, designing them to be pH-sensitive to avoid degradation throughout their bodily journey. Both in vitro and in vivo results underscore the NPs' effectiveness in transporting the anti-miR, with thorough particle characterization and statistically significant results. This work, offering an advancement in the drug delivery field, warrants publication but would benefit from an author review to add crucial information, ensuring reproducibility of the results by others in the future.

Specific comments:

In Materials and methods:

·      PVA does not contain MW.

·      Authors should include the solvents in which the polymers were dissolved. For example, what solvent was chitosan? Di or PBS? In what solvent was the NHS dissolved and what pH. NHS is poorly soluble in water. Also, concentrations of EDC NHS are not reported.

In general, the authors should review their methods section for clarity.

Author Response

Review 2

1) PVA does not contain MW.

Answer: We gratefully thank the reviewer for the great advice. We have added the molecular weight of PVA (MW 74.8-79.2 kDa) in the manuscripts.

2) Authors should include the solvents in which the polymers were dissolved. For example, what solvent was chitosan? Di or PBS? In what solvent was the NHS dissolved and what pH. NHS is poorly soluble in water. Also, concentrations of EDC NHS are not reported.

Answer: We gratefully thank the reviewer for the great advice. The CS solvent was acetic acid solution with pH at 4.5. The solubility of N-hydroxysuccinimide (NHS), which purchased from Damas, is 50 mg/mL (great soluble in water). The concentrations of 1-ethyl-3-(3-dimethylaminopropyl) carbodiimide (EDC) and NHS is 6.0 mg/mL and 6.5 mg/mL, respectively. The above information has been added in the manuscripts according to your suggestion.

3) In general, the authors should review their methods section for clarity.

Answer: We gratefully thank the reviewer for the great advice. We have checked and revised the methods section based on your suggestions.

Reviewer 3 Report

Review report

In the article titled: “pH-sensitive nanoparticles for colonic delivery anti-miR-301a in mouse models of inflammatory bowel diseases” is investigated the possibility of using multifunctional oral nanoparticle delivery system (pH-sensitive HA-CS/ES100/PLGA nanoparticles) loaded with anti-miR for improving the targeting ability and the therapeutic efficacy in inflammatory bowel disease. The authors studied cytotoxicity, cellular uptake,

 pharmacodynamic properties and  reduction of pro-inflammatory factors by nanoparticle delivery system.

Materials and methods

- 2.2. Preparation of nanoparticles

Please, provide information of “tris solution” (line 147). Guess should be a buffer and should be provide the pH of the buffer.

- 2.8. Cytotoxicity assay and 2.9. Cellular uptake and 2.10. In vitro cytokine assay

Please, in 4 × 104 cells (line 202) 4 should be in superscript and 1 × 105 cells (line 211) 5 –superscript.

The same for 4 × 104 cells (line 221).

-3.2. Storage Stability

In order to justify the storage stability of the nanoparticles authors claimed that “the concentration of the FAM-anti- miR remained a slight variation during the week, indicating the superior stability of loading anti-miR and that the anti-miR@HCeP NP showed that nanoparticles could compress the gene effectively at 1, 4 and 7 d”. The authors should describe the conditions (as pH) which are used during the above experiments in which the storage stability was proven. It will explain the differences in the pH conditions (adjusted later in the next section 3.4. In vitro drug release  when the drug is already release effectively as a function of pH.

-2.8. Cytotoxicity assay

              For cytotoxity assay should be tested and FAM-anti-miR@HCeP NP, since they will be used later in 2.9. Cellular uptake experiment

anti-miR@HCeP NP and show good stability, no cytotoxicity, acid resistance and pH response. In addition, nanoparticles demonstrated good penetrating ability in Caco-2 cells.

Using DSS-induced IBD mouse models authors demonstrated excellent therapeutic effects on DAI, Morris score, colon length, levels of several pro-inflammatory factors, and histopathology via the anti-miR@HCeP NP.

My opinion is that the article presents interesting results regarding the evaluation of a new drug release system and can be published after minor revision.

Author Response

Review 3

1) Materials and methods

- 2.2. Preparation of nanoparticles

Please, provide information of “tris solution” (line 147). Guess should be a buffer and should be provide the pH of the buffer.

Answer: We gratefully thank the reviewer for the great advice. The pH of tris solution is 7.0. The above information has been added in the manuscripts according to your suggestion.

2) - 2.8. Cytotoxicity assay and 2.9. Cellular uptake and 2.10. In vitro cytokine assay

Please, in 4 × 104 cells (line 202) 4 should be in superscript and 1 × 105 cells (line 211) 5 –superscript.

The same for 4 × 104 cells (line 221).

Answer: We gratefully thank the reviewer for the great advice. The above information has been added in the manuscripts according to your suggestion again.

3) -3.2. Storage Stability

In order to justify the storage stability of the nanoparticles authors claimed that “the concentration of the FAM-anti-miR remained a slight variation during the week, indicating the superior stability of loading anti-miR and that the anti-miR@HCeP NP showed that nanoparticles could compress the gene effectively at 1, 4 and 7 d”. The authors should describe the conditions (as pH) which are used during the above experiments in which the storage stability was proven. It will explain the differences in the pH conditions (adjusted later in the next section 3.4. In vitro drug release when the drug is already release effectively as a function of pH.

Answer: We gratefully thank the reviewer for the great advice. The anti-miR@HCeP NP was stored at 4 ℃ (pH 6.5). The above information has been added in the manuscripts according to your suggestion.

4) -2.8. Cytotoxicity assay

For cytotoxity assay should be tested and FAM-anti-miR@HCeP NP, since they will be used later in 2.9. Cellular uptake experiment

Answer: Really appreciate for your professional and pertinent comments. As shown in Figure 2(A), the cell viability of HCeP NP remained acceptable even at 500 μg/mL. Additionally, the cell viability of anti-miR@HCeP NP remained acceptable even at 30 nM (anti-miR-301a concentration). To note, FAM labeled gene has been widely used in gene uptake studies [1]. Next, this paper evaluated the ability of Caco-2 and raw 264.7 cells to uptake FAM-anti-miR@HCeP NP at 20 nM (anti-miR-301a concentration). Therefore, we believe that FAM-anti-miR@HCeP NP is safe for further study. Also, we conducted a safety assessment about FAM-anti-miR@HCeP NP in Figure S2. Therefore, the subsequent cytological investigations, including cellular uptake experiment, were carried out at safe concentrations.

References

  1. Huang, X.; Schwind, S.; Yu, B.; Santhanam, R.; Wang, H.; Hoellerbauer, P.; Mims, A.; Klisovic, R.; Walker, A.R.; Chan, K.K.; et al. Targeted delivery of microRNA-29b by transferrin-conjugated anionic lipopolyplex nanoparticles: a novel therapeutic strategy in acute myeloid leukemia. Clin Cancer Res 2013, 19, 2355–2367.